# Remote and Unsupervised Exercise Strategies for Improving the Physical Activity of Colorectal Cancer Patients: A Meta-Analysis

**DOI:** 10.3390/healthcare11050723

**Published:** 2023-03-01

**Authors:** Andrea Corazzi Pelosi, Gabriela Cerávolo Rostirola, Juliana Silva Pereira, Karen Christine Silva, Maria Eduarda Ramos Fontanari, Manoela Stahl Parisotto Oliveira, Ivan Gustavo Masselli dos Reis, Leonardo Henrique Dalcheco Messias

**Affiliations:** Research Group on Technology Applied to Exercise Physiology (GTAFE), Laboratory of Multidisciplinary Research, São Francisco University, Bragança Paulista 12916-900, Brazil

**Keywords:** neoplasm, physical exercise, rehabilitation, colon, cancer

## Abstract

Colorectal cancer (CRC) burden across the world is expected to increase by ~2.2 million new cases and ~1.1 million deaths by 2030. Regular physical exercise is recommended to prevent CRC, but the myriad of protocols preclude further discussion on how to manage its variables for this population. Home-based exercise guided by remote monitoring provides an alternative to surpass the barriers of supervised exercise. However, no meta-analysis was conducted to verify the effectiveness of this intervention for improving physical activity (PA). We performed a systematic review of remote and unsupervised strategies imposed on CRC patients for improving PA and compared, via a meta-analysis, their effectiveness against CRC patients submitted to usual care or no intervention. The databases PubMed, Scopus, and Web of Science were searched on 20 September 2022. Eleven studies attained the criteria for eligibility in the qualitative approach, and seven were included in the meta-analysis. No significant effect (*p* = 0.06) of remote and unsupervised exercise intervention was observed. However, a sensitivity analysis including three studies that only considered CRC patients was performed, demonstrating a significant effect in favor of exercise (*p* = 0.008). Based on our sensitivity analysis, remote and unsupervised exercise strategies were effective to improve the PA of CRC patients.

## 1. Introduction

Healthy behaviors, like engaging in physical exercise, benefit colorectal cancer (CRC) patients by improving their quality of life [1,2,3,4,5,6]. Previous meta-analyses have focused on the benefits of physical exercise for CRC patients, including improvements in aerobic power and metabolism [7], quality of life, and functional capacity [8], in addition to being feasible [9]. Further, a large body of scientific data has concluded that physical activity (PA) in distinct ages or conditions can improve, even during the treatment or in long-term follow-up, both the survival rates and mental health of CRC patients [3,4,10,11,12,13,14,15]. In this scenario, new strategies to improve the PA in this population are paramount.

Programs to improve PA can be trialed in hospital settings, especially for a patient awaiting surgery. However, hospital-based initiatives may not be suitable for everyone due to other commitments, travel difficulties, distance and costs, multi-morbidity, and discomfort in group settings. Therefore, health care can be provided at home, mainly for immunocompromised patients at risk of infections [16]. In this sense, home-based interventions could be a promising alternative to hospital-based rehabilitation. To assess compliance, strategies based on a daily diary or logbook are valid to record all activities, besides the possibility of remote monitoring via telephone calls, for instance [17,18]. 

Exercise practice guided by new technologies had great momentum during the pandemic period [16] and was strongly recommended to maintain or improve the PA of cancer patients, regardless of the cancer type [19]. However, diverse strategies with the same goal have been applied well before the coronavirus outbreak. Approaches using health coaching delivered by telephone, internet, or combined methods were associated with improved quality of life, mood, and PA of patients with any type of cancer, but not self-efficacy [20]. Regarding CRC patients, most studies in a systematic review reported a positive effect of using eHealth, a system that incorporates a wide range of applications facilitating communication between patients and healthcare professionals and monitoring patients’ health status, providing useful services for supporting this population [21]. On the other hand, delivering a digital versatile disk (DVD) containing exercises or pedometers recorded monthly showed no improvement in the quality of life of these patients [22].

Overall, controversial data is available regarding the effectiveness of remote and non-presential supervision exercise strategies for CRC patients, including the improvements in PA. In this context, a myriad of strategies based on workbooks, booklets, newsletters, motivational interviews, and telephone counseling was applied in the specialized literature. However, the scientific community lacks some systematic review along with a meta-analysis to explore these approaches and verify their efficiency to improve the PA of CRC patients. Since regular physical exercise has been associated with CRC-specific and all-cause mortality [23], gathering information on these approaches as well as their benefits on PA becomes relevant. Therefore, we performed a systematic review of remote and unsupervised strategies imposed on CRC patients for improving PA and compared, via a meta-analysis, their effectiveness against CRC patients submitted to usual care or no intervention. Given the importance of physical exercise for this population and the available technological and remote resources to increase PA, we hypothesized that these strategies will improve this outcome of CRC patients.

## 2. Materials and Methods

### 2.1. Search Strategy

The databases PubMed, Scopus, and Web of Science were searched on 20 September 2022. One researcher (LHDM) with experience in advanced searches retrieved the studies for the initial screening. The Boolean operators “AND” and “OR” were used along with the terms “physical activity OR physical exercise OR physical training” AND “colorectal neoplasm OR colorectal cancer”. The “English language” and “human” filters were activated whenever possible. The titles were transposed to a Microsoft Excel datasheet with three tabs, one per database. The screening supervisor (LHDM) checked all titles and excluded duplicates or triplicates manually. Subsequently, four of these Excel files were generated, each one with the name of the reviewer. This procedure secured the blind revision.

### 2.2. Eligibility Criteria

The screening was performed based on the Preferred Reporting Items for Systematic Reviews and Meta-Analyses (PRISMA) [24]. Furthermore, the Population (P), Intervention (I), Comparator (C), Outcomes (O), and Study design (S) (PICOS) [25] structure guided reviewers to include or exclude the studies. Therefore, the PICOS consisted of CRC patients (P) submitted to remote and unsupervised exercise (I) and compared to CRC patients without exercise (C) to verify the intervention’s effect on the PA levels (O) in randomized controlled trials (S).

The meticulous inclusion consisted of: (i) studies showing the effects of the intervention of PA levels rather than only associations between this parameter with other outcomes; (ii) interventions without presential supervision, but that included some presential visits through the intervention for resolving doubts of asses the outcomes, including the PA; (iii) interventions with some level of follow-up during the period, including phone calls, instruction books, pamphlets, DVDs, letters, home logs, trackers, or any way to instruct the CRC patients; (iv) the length of the intervention was clearly presented; (v) the remote and unsupervised strategy was clearly presented, allowing reproduction; (vi) some PA outcome was measured, including questionnaires, pedometers or other equipment that retrieved this parameter; and (vii) studies in English. Manuscripts were not included if: (i) only the abstract was available; (ii) no CRC patient was included; (iii) mean or variance was not provided; (iv) missed control or usual care group, (v) were transversal intervention, and (vi) provided supervision during the physical exercise.

### 2.3. Data Extraction and Quality Assessment

Four reviewers (GCR, JSP, MERF, and MSPO) independently screened the studies. In each screening round (i.e., title, abstract, and full text) another researcher (LHDM) retrieved the Excel sheets and verified the discrepancies. The same researcher brought the reviewers together to discuss disagreements, and a resolution was reached in all cases. Information on the sample size, age, type and stage of cancer, intervention strategy, and PA results were retrieved. Another researcher (KCS) double-checked the information retrieved. The corresponding authors were contacted when the information presented in the studies was insufficient or inconclusive. If no response was provided, the study was not included. The quality assessment was performed by one reviewer (LHDM) using the Physiotherapy Evidence Database (PEDro) scale [26]. Studies were classified from fair to substantial or fair to good [27].

### 2.4. Meta-Analysis

The PA (dependent variable) retrieved from seven studies [28,29,30,31,32,33,34] was considered a continuous variable. Four studies [35,36,37,38] were not included since no variance or mean was provided. The Review Manager 5 software (version 5.3, The Nordic Cochrane Centre, Copenhagen, Denmark) was used for the meta-analysis. A random effects model was used for the analysis and post-intervention means and standard deviations (SDs) were adopted for comparisons. The standardized mean difference (SMD) was considered as the effect size index along with 95% confidence intervals (CI). Significance analysis, proportion estimation of variance, and variation in the treatment effects were based on the Z-value, I2, and prediction interval (PI), respectively. The PI was calculated by the comprehensive meta-analysis software (CMA) [39]. The same software was used to calculate Egger’s regression intercept and the Fail-safe N. Since several studies included in the qualitative search also considered patients with other cancers, we performed a sensitivity analysis with those that only considered CRC patients. The statistical significance was set at *p* < 0.05.

## 3. Results

The initial search retrieved 1535, of which 11 attained the criteria for eligibility in the qualitative approach, andseven provided enough data to be included and the meta-analysis (Figure 1). The studies were published between 2009 and 2019. The PEDro score was 7.27 ± 0.90, with three studies classified as six, two with seven, and the remaining attained eight (Table A1).

Information on sample size, age, type, and stage of cancer was synthesized in Table 1. A total of 2805 cancer patients were enrolled in randomized controlled trials, of which 1405 initiated remote and unsupervised exercise intervention and 1400 received usual care or no intervention. The raw comparison showed that male cancer patients were higher (1462 interventions = 735; control = 727) than females (1343 interventions = 670; control = 673) in the overall sample. Exclusively for CRC patients, 682 were randomized in the control or usual care group, while 672 initiated the intervention. Similar age was observed for both exercised and controls (63 ± 7 years in both groups). Six studies [28,29,32,34,35,37] considered patients with cancers more than CRC, while the remaining five [30,31,33,36,38] only considered patients with this neoplasm. 

The remote and unsupervised exercise strategies, the instruments for measuring PA, and the results of the trials are shown in Table 2. Studies presented large heterogeneity regarding the intervention, including workbooks, letters, phone calls, health coach sessions, trackers, and distinct motivational approaches. Likewise, different questionnaires were adopted in eight studies [28,29,30,32,33,34,37,38] for measuring the PA, while accelerometers retrieved this outcome in three randomized controlled trials [31,35,36]. The mean length of interventions was 31 ± 26 weeks. Short interventions (four weeks) were observed in only one study [37], while others submitted cancer patients to 12 [35,36,38], 16 [32], 24 [29,33], 48 [28,30,34], and 96 [31] weeks of physical exercise or follow-up.

The meta-analysis including 1262 cancer patients and 1259 controls showed no significant effect of remote and unsupervised exercise intervention for promoting PA improvement (Figure 2A). Additionally, a high proportion of variance was observed along with PI between −3.12 to 5.36. Egger’s regression intercept was observed at 2.02 and a *p*-value of 0.46. Regarding the Fail-safe N, Z-value was obtained at 23.3 with a *p*-value of 0.000, and the number of missing studies that would bring the *p*-value to > alpha was 983. The sensitivity analysis including studies that only considered CRC patients (i.e., 354 patients and 351 controls) demonstrated a significant effect in favor of exercise (Figure 2B). Further, the CMA returned a common effect size.

## 4. Discussion

This meta-analysis demonstrated large variability in terms of sample size, interventions, and instruments to measure PA in remote and unsupervised exercise interventions for cancer patients. These factors may have contributed to the non-significant effect of the intervention. On the other hand, our sensitivity analysis suggests that exercise interventions with remote monitoring can improve the PA of CRC patients.

The better approach regarding exercise prescription for CRC patients—as for other neoplasms—are in course of discussion. Supervised exercise training improves the functional capacity following colorectal cancer resection, also promoting faster recovery to regular patients’ activities [41]. Using a qualitative approach, Hatlevoll et al., [42] concluded that presential appointments with a physiotherapist serve as an important external motivational factor. Further, the CRC patients in this study reported improvements in muscle strength and mental health, along with a reduction in sensory neuropathic symptoms.

Despite the advantages of supervised exercise, patients with cancer may face barriers that limit their adherence to this intervention, including, for instance, time and cost [40]. On the other hand, a myriad of approaches and definitions of the “home-based” setting is available [43], leading to distinct conclusions regarding this approach. Such a perspective was also observed in this study. From the seven studies included in the meta-analysis, five [29,30,31,32,33] reported a large variance in PA after the exercise program. Another factor that may contribute to the non-significant effect of this analysis is the usual care adopted in many of the included studies. Although it is merely speculative, the fact that some interventions also included motivational implements or workbooks with guidance for healthy behaviors may have improved the PA of controls. On the other hand, these approaches are fully expected, since it is not ethically acceptable to prevent control patients from receiving materials to improve their health.

Both supervised and home-based approaches have advantages and disadvantages. However, a recent meta-analysis by our group has suggested that adherence to the intervention, regardless of the supervision level, is relevant for improving the functional capacity and quality of life of CRC patients [8]. In this scenario, new technologies and instruments are valid alternatives to inform, motivate, and remember the PA benefits for these patients, ultimately improving their intervention adherence. Such a suggestion is supported by the findings of Fisher et al., [44], who observed that recall of PA advice after a diagnosis of CRC was associated with higher levels of this outcome.

The systematic review by Wang et al., [5] of behavioral interventions using the web and mobile technology enrolled patients with several kinds of cancer, among them, CRC. These patients were submitted to mobile health (mHealth), defined by the World Health Organization (WHO) as medical and public health practices supported by a mobile device, such as a mobile phone, patient monitoring devices, personal digital assistants, and other wireless devices. The results show that mHealth interventions are a promising approach to improving PA and dietary behaviors in cancer survivors.

Despite the promising results, this meta-analysis must be comprehended along with its limitations. Specifically for the included studies, it is possible to observe a lack of standardization regarding the period of interventions. This prevents a conclusion about the minimum time needed to observe an increase in PA in the studied population. A similar limitation resides in the format of the interventions. Although all are part of remote and unsupervised interventions, each study used a different strategy. Additionally, only three studies presented specific results for CRC patients, which is a limiting factor for increasing the power of the meta-analysis.

Based on our eligibility criteria, a few studies were inserted in both qualitative and quantitative analysis. The main issue in this context is the absence of PA parameters in randomized and controlled trials. This factor accounted for 46% of the reasons for exclusion. Further, the heterogeneity observed precludes deep interpretations regarding the effectiveness of remote and unsupervised exercise interventions for improving the PA of patients with cancer, including CRC. In this sense, we opted to include studies with other neoplasms than CRC to strengthen the qualitative analysis, also improving the understanding of remote and unsupervised exercise strategies for CRC patients. To solve this limitation, future studies must present the separated results according to the cancer type. Apart from the limitations, we could perform the sensitivity analysis and explore the effects of these interventions to improve the PA of CRC patients. This factor can be comprehended as the main strength and innovation of this meta-analysis.

## 5. Conclusions

This meta-analysis verified the effects of remote and unsupervised exercise strategies on the PA of CRC patients. When the analysis was performed with studies that considered other cancers, no significant effect of the exercise intervention was observed, along with large heterogeneity. However, based on our sensitivity analysis, remote and unsupervised exercise strategies were effective in improving the PA of CRC patients. This result demonstrated the relevance of PA for this population, promoting the benefits derived from physical exercise and acting as an adjuvant in the treatment of this disease.

## Figures and Tables

**Figure 1 healthcare-11-00723-f001:**
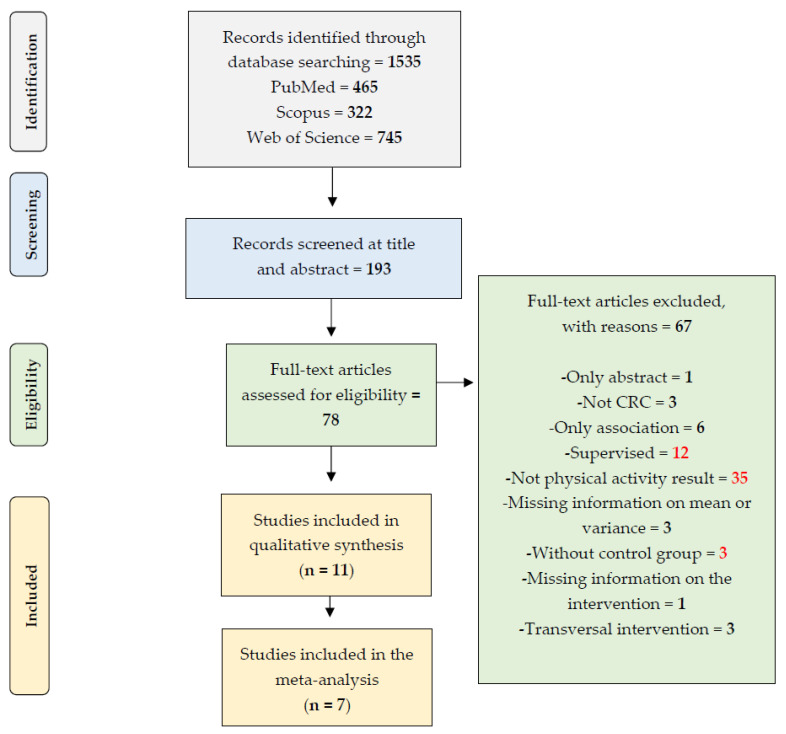
The Preferred Reporting Items for Systematic Reviews and Meta-Analyses (PRISMA) used to guide the systematic review.

**Figure 2 healthcare-11-00723-f002:**
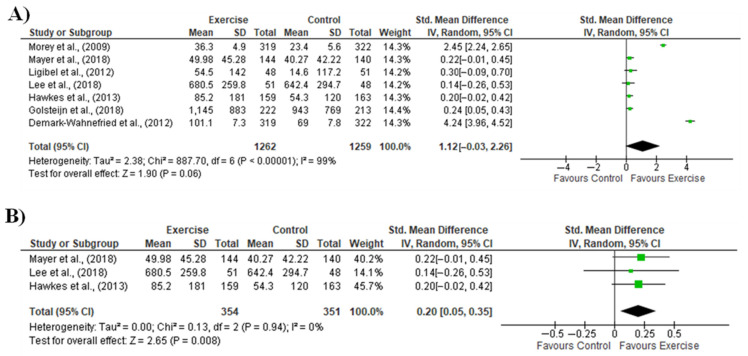
Meta-analysis of physical activity (dependent variable) measured in randomized controlled trials that conducted remote and unsupervised exercise for cancer patients; (**A**) Comparison of the physical activity levels between cancer patients—including colorectal—submitted to remote and unsupervised exercise and cancer patients in the control group; (**B**) Sensitivity analysis considering the studies that only included patients diagnosed with colorectal cancer [28,29,30,31,32,33,34].

**Table 1 healthcare-11-00723-t001:** Summary of the included studies in the qualitative synthesis.

Study	Sample Size	Age	Only CRC Patients? Stage of Cancer or Condition
Morey et al., (2009) [34]	*Control*—*n =* 322 Female—*n =* 177 Male—*n =* 145 *Intervention*—*n =* 319 Female—*n =* 172 Male—*n =* 147	*Control* = 73 ± 5 yrs *Intervention* = 73 ± 5 yrs	No *Control* Breast—*n =* 146 Prostate—*n =* 130 Colorectal—*n =* 46 Years since cancer diagnosis = 8 ± 2 *Intervention* Breast—*n =* 143 Prostate—*n =* 131 Colorectal—*n =* 45 Years since cancer diagnosis = 8 ± 2
Ligibel et al., (2012) [32]	*Control*—*n =* 60 Female—*n =* 56 Male—*n =* 4 *Intervention*—*n =* 61 Female—*n =* 56 Male—*n =* 5	*Control* = 55 ± 10 yrs *Intervention* = 53 ± 10 yrs	No *Control* Breast—*n =* 50 Colorectal—*n =* 10 Stage 1 = 21; Stage 2 = 23; Stage 3 = 16 *Intervention* Breast—*n =* 50 Colorectal—*n =* 11 Stage 1 = 20; Stage 2 = 19; Stage 3 = 22
^#^ Demark-Wahnefried et al., (2012) [28]	*Control*—*n =* 245 Female—*n =* 138 Male—*n =* 107 *Intervention*—*n =* 243 Female—*n =* 132 Male—*n =* 111	*Control* = 72 ± 5 yrs *Intervention* = 73 ± 5 yrs	No *Control* Breast—*n =* 110 Prostate—*n =* 94 Colorectal—*n =* 41 *Intervention* Breast—*n =* 111 Prostate—*n =* 99 Colorectal—*n =* 33 ≥5 years from diagnosis
Hawkes et al., (2013) [30]	*Control*—*n =* 205 Female—*n =* 90 Male—*n =* 115 *Intervention*—*n =* 205 Female—*n =* 99 Male—*n =* 106	*Control* = 67 ± 9 yrs *Intervention* = 64 ± 10 yrs	Yes *Control*—Dukes’ staging—A = 39; B = 53; C = 48; Unknown = 65; *Intervention*—Dukes’ staging—A = 36; B = 65; C = 45; Unknown = 59
Pinto et al., (2013) [38]	*Control*—*n =* 26 Female—*n =* 14 Male—*n =* 12 *Intervention*—*n =* 20 Female—*n =* 12 Male—*n =* 8	*Control* = 55 ± 8 yrs *Intervention* = 59 ± 11 yrs	Yes Stages 1, 2 or 3
Park et al., (2015) [37]	*Control*—*n =* 59 Female—*n =* 50 Male—*n =* 9 *Intervention*—*n =* 50 Female—*n =* 45 Male—*n =* 5	*Control* = 53 ± 8 yrs *Intervention* = 50 ± 8 yrs	No *Control* Breast—*n =* 41 Colorectal—*n =* 18 Stage 1 = 22; Stage 2 = 21; Stage 3 = 12 *Intervention* Breast—*n =* 39 Colorectal—*n =* 11 Stage 1 = 19; Stage 2 = 15; Stage 3 = 8
Mayer et al., (2018) [33]	*Control*—*n =* 140 Female—*n =* 73 Male—*n =* 67 *Intervention*—*n =* 144 Female—*n =* 74 Male—*n =* 70	*Control* = 57 ± 14 yrs (*n* = 104) *Intervention* = 59 ± 13 yrs (*n* = 115)	Yes *Control*—Stage 1 = 27; Stage 2 = 82; Stage 3 = 29 *Intervention*—Stage 1 = 39; Stage 2 = 63; Stage 3 = 41
Lee et al., (2018) [31]	*Control*—*n* = 56 Female—*n =* 26 Male—*n =* 30 *Intervention*—*n =* 56 Female—*n =* 16 Male—*n =* 40	*Control* = 64 ± 9 yrs *Intervention* =66 ± 9 yrs	Yes *Control*—Stage 1 = 12; Stage 2 = 24; Stage 3 or 4= 20 *Intervention*—Stage 1 = 8; Stage 2 = 27; Stage 3 or 4= 20
Golsteijn et al., (2018) [29]	*Control*—*n =* 229 Female—*n =* 25 Male—*n =* 204 *Intervention*—*n =* 249 Female—*n =* 37 Male—*n =* 212	*Control* = 66 ± 8 yrs *Intervention* = 66 ± 7 yrs	No *Control* Prostate—*n =* 143 Colorectal—*n =* 86 *Intervention* Prostate—*n =* 149 Colorectal—*n =* 100 At least 6 weeks post-surgery
Maxwell-Smith et al., (2018) [40]	*Control*—*n =* 34 Female—*n =* 13 Male—*n =* 21 *Intervention*—*n =* 34 Female—*n =* 21 Male—*n =* 13	*Control* = 62 ± 8 yrs *Intervention* = 65 ± 7 yrs	No *Control* Gynaecologic—*n =* 4 Colorectal—*n =* 30 *Intervention* Gynaecologic—*n =* 11 Colorectal—*n =* 23 Stages 1 or 2
Moug et al., (2019) [36]	*Control*—*n =* 24 Female—*n* = 11 Male—*n =* 13 *Intervention*—*n =* 24 Female—*n =* 6 Male—*n =* 18	*Control* = 66 ± 9 yrs *Intervention* = 65 ± 11 yrs	Yes New diagnosis

The “*n*” inserted in the table refers to the initial. Some studies only reported the participant’s status at the baseline of the total “n”, but did not present the characteristic of the final “n”.

**Table 2 healthcare-11-00723-t002:** Intervention strategy, length, and the results of the included studies in the qualitative synthesis.

Study	-Intervention Strategy -Control -Length	Physical Activity Parameter (Dependent Variable) and Results
Morey et al., (2009) [34]	- RENEW: personally tailored workbook and series of quarterly newsletters, along with a program of telephone counseling and automated prompts (i.e., 15 sessions and 8 prompts over the 12-month period); - A delayed intervention, wait-list control was used; - 48 weeks.	Community Health Activities Model Program for Seniors questionnaire ^§^ Control Baseline = 28.7 ± 2.3 min/wk 48 wk = 23.4 ± 5.6 min/wk Intervention Baseline = 24.6 ± 2.1 min/wk 48 wk = 36.3 ± 4.9 min/wk
Ligibel et al., (2012) [32]	- 10–11 semi-structured phone calls. The initial calls focused on goal setting and performance assessment so as to build self-efficacy for exercise behaviors, while later calls concentrated on the adequacy of plans for relapse prevention calls; - The control group received routine care during the 16 weeks and was then offered a telephone consultation with an exercise trainer at the end of the period; - 16 weeks.	7-Day Physical Activity Recall Control Baseline = 65.7 ± 84.1 min/wk 16 wk = 14.6 ± 117.2 min/wk Intervention Baseline = 44.9 ± 58.5 min/wk 16 wk = 54.5 ± 142.0 min/wk
Demark-Wahnefried et al., (2012) [28]	- RENEW: personally tailored workbook and series of quarterly newsletters, along with a program of telephone counseling and automated prompts (i.e., 15 sessions and 8 prompts over the 12-month period); - A delayed intervention, wait-list control was used; - 48 weeks.	Community Health Activities Model Program for Seniors questionnaire Control Baseline = 37.5 ± 3.2 min/wk 48 wk = 69.0 ± 7.8 min/wk Intervention Baseline = 33.3 ± 2.9 min/wk 48 wk = 101.1 ± 7.3 min/wk
Hawkes et al., (2013) [30]	- 11 telephone-delivered health coaching sessions over a 6-month period (biweekly for 5 months, followed by a final telephone session 4 weeks later to promote self-management techniques and maintenance of behavioral improvements), a participant handbook, regular motivational postcard prompts, a pedometer, and a quarterly study. The intervention occurred in the first 24 weeks, but patients were followed until 48 weeks; - The control group received four freely available educational brochures produced on understanding CRC and cutting cancer risk, diet, and physical activity. Participants also received a quarterly study newsletter to enhance participant retention and were contacted for all follow-up assessments; - 48 weeks.	Godin Leisure-Time Exercise Questionnaire Control Baseline = 52.0 ± 112.5 min/wk 48 wk = 54.3 ± 120.0 min/wk Intervention Baseline = 58.9 ± 132.9 min/wk 48 wk = 85.2 ± 181.0 min/wk
Pinto et al., (2013) [38]	- Participants received in-person instructions on how to exercise at a moderate intensity level, monitor heart rate, and warm up before and cool down after exercise. They were also given home logs to monitor physical activity participation and a pedometer. Lastly, each participant received a weekly call over 12 weeks from research staff to monitor physical activity participation, identify relevant health problems, problem-solve any barriers to physical activity, and reinforce participants for their efforts; - The control group received weekly calls over 12 weeks. The group also received CRC survivorship tip sheets; - 12 weeks.	Seven-day Physical Activity Recall Control ^§§^ Baseline = 30 min/wk 12 wk = 146 min/wk Intervention ^§§^ Baseline = 30 min/wk 12 wk = 88 min/wk
Park et al., (2015) [37]	- Oncologist’s exercise recommendation for the practice of physical exercise combined with an exercise motivation package group. The exercise motivation package included exercise—DVDs, a pedometer, an exercise diary, and a 15-min exercise education session; - The control group received the conventional treatment consultation but received no exercise recommendation; - 4 weeks.	Godin Leisure-Time Exercise Questionnaire Control ^§§^ Baseline = 254.56 min/wk 4 wk = −39.05 min/wk Intervention ^§§^ Baseline = 187.85 min/wk 4 wk = +47.57 min/wk
Mayer et al., (2018) [33]	- Participants received National Cancer Institute’s “Facing Forward: Life after Cancer Treatment” booklet, the National Coalition for Cancer Survivorship’s Cancer Survival Toolbox, a pedometer, smartphones with the SurvivorCHESS application, along with voice and data services for the study period; - The control group received the same items, except for the SurvivorCHESS application; - 24 weeks.	Godin Leisure-Time Exercise Questionnaire Control Baseline = 15.49 ± 27.6 min/wk 24 wk = 40.27 ± 42.22 min/wk Intervention Baseline = 19.43 ± 27.07 min/wk 24 wk = 49.98 ± 45.28 min/wk
Lee et al., (2018) [31]	- One session of face-to-face motivational interview, fortnightly motivational phone calls, mailed monthly stage-of-change matched educational pamphlets, mailed quarterly newsletters, and quarterly group meetings. The intervention occurred in the first 48 weeks but was followed for 96 weeks; - The control group received five pamphlets containing general health advice; - 96 weeks.	Accelerometer ^§§§^ Control Baseline = 473.3 ± 267.3 min/wk 96 wk = 642.4 ± 294.7 min/wk Intervention Baseline = 460.8 ± 239.6 min/wk 96 wk = 680.5 ± 259.8 min/wk
Golsteijn et al., (2018) [29]	- OncoActive: a computer-tailored physical activity program providing physical activity advice online and with printed materials. Participants received automatically generated personalized feedback regarding physical activity and psychosocial determinants of physical activity. Every participant received a pedometer and access to interactive content on the website, including role model videos, home exercise instruction videos, a module for goal setting using a pedometer, the option to consult a physical therapist, and additional information; - The control group received the OncoActive intervention after completing the last measurement; - 24 weeks.	Short Questionnaire to Assess Health Enhancing Physical Activity Control Baseline = 873 ± 764 min/wk 24 wk = 943 ± 769 min/wk Intervention Baseline = 780 ± 721 min/wk 24 wk = 1145 ± 883 min/wk
Maxwell-Smith et al., (2018) [40]	- WATAAP: (I) a wrist-worn tracker (Fitbit Alta) to record daily steps and distance; (II) 2-h group sessions for general information on behavior, goals, and the application; (III) 20-min phone call during week 8 to provide support and feedback regarding physical activity progress, review goals, action plans, and coping-planning strategies; - The control group received print materials containing physical activity guidelines (also given to the intervention group) but was not specifically encouraged to increase their physical activity; - 12 weeks.	Accelerometer Control ^§§^ Baseline = 158 min/wk 12 wk = 138 min/wk Intervention ^§§^ Baseline = 170 min/wk 12 wk = 186 min/wk
Moug et al., (2019) [36]	- Each participant was given a weekly walking diary (targets and motivational material included) and the use of the pedometer was explained. Participants then received follow-up telephone calls (weeks 1, 3, 5, 7, 9, 12, 16) where new stepping targets were set, motivational techniques were applied and any issues were discussed. All participants were asked to engage a support person (e.g., spouse) to assist in their adherence to the program; - The control group received standard care with no contact from the trial team except at the two test sessions and were offered a voluntary exercise counseling session and information pack from the trial team after their surgery and on completion of the trial; - 12 weeks.	Accelerometer Control Baseline = 7773 ± 3975 median steps/day 12 wk = 5920 ± 3152 median steps/day Intervention Baseline = 7779 ± 4045 steps/day ^§§§§^ 12 wk = 6675 ± 3100 steps/day ^§§§§^

^§^ The retrieved data refers to the duration of the endurance exercise; ^§§^ The variance was not presented. ^§§§^ The data refers to groups C and D, considering only the baseline and 12-month results. ^§§§§^ Median. RENEW—Reach out to Enhance Wellness. CHESS—Comprehensive Health Enhancement Support System. WATAAP—Wearable Activity Technology and Action Planning.

## Data Availability

Not applicable.

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
