# Peer review of "Remote and Unsupervised Exercise Strategies for Improving the Physical Activity of Colorectal Cancer Patients: A Meta-Analysis"

_healthcare, 2023, doi:10.3390/healthcare11050723_

Round 1

Reviewer 1 Report

The authors provide an interesting systematic review of physical activity interventions in CRC patients.

1. Why did the authors exclude pilot studies?

2. Can the authors describe which Mesh Terms were used to identify different terms?

Reviewer 2 Report

The article is well structured and meets the quality requirements for publication in its current form.

Its introduction is in line with the subject matter of the systematic review and makes use of current bibliographical references in accordance with the subject matter of the article.

The method is perfectly defined and the PRISMA methodology is detailed and is in accordance with this type of work. The existence of four external reviewers gives validity to this process.

The results are presented correctly, both from the systematic review and the meta-analysis, and the tables are easy for the reader to interpret.

The discussion and conclusions are also correct and in line with the results that have been presented previously.

In short, I believe that the authors have carried out a systematic review and meta-analysis of a topic of great interest for the health of colorectal cancer patients and the influence of physical activity on the disease in a correct manner.

Reviewer 3 Report

Congratulations on a very interesting article in terms of subject matter and generally well focused. Here are some suggestions for improvement:

1) There is a problem that applies from the beginning. You talk about "effects" but at no point is it clearly specified which is the dependent variable(s) being analysed in this study. This requires a global approach throughout the study.

2) In the introduction in many sections you talk about cancer in a generalised way, but without specifying what types of cancer you are talking about. Please correct this or justify that the use of physical exercise as a health improvement measure in cancer patients could be general, regardless of the type of cancer.

3) The introduction does not clearly state the gaps that exist in the previous literature for this research to be necessary. This is an issue that needs to be addressed.

4) Please include the hypothesis of your research.

5) Was the review protocol recorded? This should be specified.

6) Please include the choice of English language as an inclusion criterion for the article.

7) Please specify what you considered a "pilot study" and on what basis to establish exclusion criteria.

8) Please calculate Cohen's Kappa values and include them in the study.

9) Please calculate Egger's bias statistics and Rosenthal's fail-safe N.

10) Include a scale to analyse the quality of the articles included in the review and discuss the results found.

11) Given that you have searched three different databases, how do you avoid duplicate or triplicate articles?

12) The criteria specified in Figure 1 do not match the inclusion and exclusion criteria. Please correct this.

13) Include in any of the tables clearly the dependent variables being assessed, as well as the units in which they are assessed and the assessment instruments used.

14) What is the dependent variable in figure 2? This is not being made clear.

15) Include a paragraph with the limitations of the studies included in the review and the future lines of research to be addressed in this respect.

Round 2

Reviewer 3 Report

Thank you for the excellent work you have done on this review. I have just a few small questions:
- It is usual for an investigation of this type to make a registration in PROSPERO. If you have done so, please include the registration code. If not, please explain why.
- On the issue of the pilot study, if these studies entered the screening, then this should be specified in the text, not being an inclusion or exclusion criterion, as it has not been the reason why these articles have been discarded.
- It is advisable to include the results of the PEDRO scale for each of the articles in a table in order to be able to track item-by-item their strengths and weaknesses.
- Information about how duplicate and triplicate items were eliminated should be included in the text.
- You need to include two different limitations paragraphs: one with the limitations of the articles that have been found in the search, and a second with the limitations of your own review. This should be made clear.
